# Separation of Simultaneous Speakers with Acoustic Vector Sensor

**DOI:** 10.3390/s25051509

**Published:** 2025-02-28

**Authors:** Józef Kotus, Grzegorz Szwoch

**Affiliations:** Department of Multimedia Systems, Faculty of Electronics, Telecommunication and Informatics, Gdansk University of Technology, 80-233 Gdansk, Poland; grzszwoc@pg.edu.pl

**Keywords:** sound intensity, sound source separation, acoustic vector sensor

## Abstract

This paper presents a method of sound source separation in live audio signals, based on sound intensity analysis. Sound pressure signals recorded with an acoustic vector sensor are analyzed, and the spectral distribution of sound intensity in two dimensions is calculated. Spectral components of the analyzed signal are selected based on the calculated source direction, which leads to a spatial filtration of the sound. The experiments were performed with test signals convolved with impulse responses of a real sensor, recorded for a varying sound source position. The experiments evaluated the proposed method’s ability to separate sound sources, depending on their position, spectral content, and signal-to-noise ratio, especially when multiple sources are active at the same time. The obtained results are presented and discussed. The proposed algorithm provided signal-to-distortion ratio (SDR) values 10–12 dB, and Short-Time Objective Intelligibility Measure (STOI) values in the range 0.86–0.94, an increase by 0.15–0.30 compared with the unprocessed speech signal. The proposed method is intended for applications in automated speech recognition systems, speaker diarization, and separation in the concurrent speech scenarios, using a small acoustic sensor.

## 1. Introduction

Automatic speech recognition (ASR) is a tool for transcription of a speech into a text. In practical scenarios, multiple speakers may be active at the same time, and the output of an ASR will be a stream of the recognized speech from all speakers. Segments of the recognized speech need to be assigned to separate speakers [1,2,3]. Two cases should be considered. If only one speaker is active at a time, the procedure of assigning the recognized text to a specific speaker is called a diarization [1]. However, if multiple speakers are active at the same time, and their speech is mixed, the problem is called a separation of acoustic sources [4,5]. In the latter case, an ASR is often unable to recognize the mixed speech, unless the signal is separated into streams, each containing a single speaker [2,6]. Assuming that the sound sources are positioned at different angles relative to the sensor recording the speech, a separation of the sound sources is possible by means of spatial filtration [7,8,9]. Additionally, various noise sources may contribute to the processed signal and deteriorate the ASR performance. Spatial filtration may perform a noise suppression, as long as the noise originates from a different direction than the speakers [4].

The scenario presented in this paper involves two concurrent speakers, with speaker A providing the speech signal of interest and speaker B being an interference. It is assumed that no other sources of acoustic disturbance are present in the scene. The goal is to obtain a clean signal from speaker A only, suitable as an input to the ASR system. The context of the work presented in this paper is related to medical institutions. In one scenario, the ASR should process the speech of a medical examiner, but not the speech of a patient. In another scenario, in an emergency room, speech from a specific person needs to be processed, while other speakers are also active at the same time. The problem of interfering speech has to be solved by a spatial filtration algorithm before the signal is passed to an ASR.

The classic approach to spatial filtration is based on multi-microphone setups [7,8,9]. Directivity of the sensor may be altered by processing the pressure signals from all microphones so that only a specific direction is amplified. Many state-of-the-art solutions are based on large multi-microphone arrays [7,8,10,11]. Such an approach is unsuitable for the scenarios presented earlier—a small size and cost-effective sensor has to be used to provide signals for the spatial filtration. Therefore, the authors choose to use a small size acoustic intensity probe for this task. While most of the known methods are based on amplitude and phase relationship in pressure signals recorded with microphone arrays [7,8,9], the proposed method analyzes the acoustic intensity signals obtained from a sensor built from closely spaced microphones [12]. From the intensity signals, the direction of arrival (DoA) of sound waves is determined, and the signal is processed to retain only components originating from a given direction [11].

The problem of speaker separation in the concurrent speech scenario has been addressed by numerous researchers, using various sensor types and different approaches. A review of the most important algorithms is presented in [13,14]. There are two main groups of such algorithms. The first group performs speaker separation in single channel signals. This approach is often based on machine learning and deep learning methods [13,14,15,16]. This group of algorithms is actively developed, as large sets of speech recordings for the algorithm training are available [17]. Such methods are usually language-dependent, and, while they are effective in separating the speech from noise, they are less efficient in case of concurrent speech. The second group of methods utilizes signals for multiple sensors; they may be based on multi-modal sensors [5] or multichannel audio signals [11,14,18]. These methods often require large sensor setups and complex signal processing procedures, but they also often have some limitations imposed. For example, Ihara et al. proposed a system in which each speaker needs a separate microphone, and two mixed speech signals do not overlap in frequency [11]. The method proposed in this paper uses a single sensor for all speakers. Zhang et al. proposed a multichannel separation method combining neural networks with a sensor array of eight microphones, placed in a virtual sphere with a radius of 7.5 cm to 12.5 cm [18]. The method proposed here uses six microphones positioned on a cube with 1 cm in side length, so the sensor is significantly smaller. Another approach by Ephrat et al. is based on a joint audio–visual method, focusing audio on a desired speaker in video, using neural networks [5]. A similar multi-modal approach was demonstrated by Rahimi et al. [19]. The method proposed here is based on an acoustic domain only. Cermak describes a method for blind speech separation based on beamforming, using three microphones spaced by 4 cm [7]. Mahdian et al. proposed a method based on auditory motivated log-sigmoid masks and maximization of the kurtosis of a separated speech [8]. McDonough et al. performed speech separation using two arrays of eight microphones each, with a 20 cm diameter [20]. Chen et al. used a time-dilated convolutional network combined with a U-net network [10].

Chen et al. presented a speaker separation method using B-format microphones, providing both the acoustic pressure and the pressure gradient signals [21]. Their processing algorithm is based on parameter estimation with an expectation-maximization method used to establish time–frequency masks for speaker separation. Pertilä applied Bayesian filtering and time–frequency masking for multiple acoustic source tracking and speech separation using circular microphone matrix [22]. A similar approach, based on a circular array of 8 and 16 microphones, was proposed by Souden et al. [23]. Xionghu et al. proposed an algorithm based on the intensity vector of the acoustic field with probabilistic time–frequency masking, using the Gaussian mixture models, whose parameters are evaluated and refined iteratively using an expectation-maximization algorithm [24]. Most of the methods mentioned above allow only for offline speech processing. Shujau et al. applied an acoustic vector sensor constructed from four microphones for the real-time separation of speech sources, using time–frequency DoA estimation [25,26]. This is similar to the approach presented in this paper, but a larger microphone setup was used to obtain the signals for analysis. Olivieri et al. proposed a real-time multichannel speech separation algorithm based on the combination of a directional representation of the sound field with a neural beamformer based on a Convolutional Neural Network [27].

In this paper, the authors adapted the separation algorithm based on masking of the signal components in the frequency domain, to the intensity signals obtained from an acoustic vector sensor—a small-size array of microphones. The approach to source separation based on the directivity of the individual spectral components is known in the literature. In the experiments described in this paper, this approach is applied to sound intensity signals calculated from the pressure signals obtained from a multichannel sensor, in a scenario in which two speakers are active concurrently. Provided that a sufficient spatial distance between the speaker is maintained, the algorithm is supposed to separate the speech of one speaker from the disturbance (the other speaker). The purpose of the experiments was to evaluate the algorithm efficiency in varying conditions (different speaker voices, speaker positions and signal-to-interference ratio values), to determine the minimal requirements needed for a satisfactory level of speaker separation, and to find the optimal values of the algorithm parameters. The algorithm performance was evaluated in terms of signal-to-distortion ratio, speech intelligibility, and speech quality. To achieve the abovementioned goals, the experiments were performed in a simulated environment. The main aim of the experiments was to test whether the proposed method may be suitable for the concurrent speech separation in ASR systems. Compared with the methods based on large sensor arrays, the presented method uses a small sensor, a cube of 1 cm side length, with six microphones, to obtain the signals for analysis. Contrary to the state-of-the-art methods based on complex signal processing, using, e.g., statistical models, the proposed algorithm is simple, as it only requires computing the intensity signals and performing a simple time–frequency analysis to determine the DoA and filter the signal components. The authors decided to focus on a classic approach based on signal processing instead of the methods based on machine learning that require large datasets for training and powerful hardware to run. The proposed method uses a small and low-cost sensor, and the processing algorithm is simple enough to be run in real time on a microcomputer with moderate processing power.

The rest of this paper is organized as follows: The algorithm for calculations of sound intensity and performing the spatial filtration is described. The simulation design, results and the obtained SNR values are presented and discussed, and this paper ends with conclusions.

## 2. Materials and Methods

A block diagram of the proposed algorithm is shown in Figure 1. The input data to the algorithm were obtained from the acoustic vector sensor (AVS), consisting of six microphones providing pressure signals (four channels are used in the analysis). In this paper, the following scenario was considered. One sound source produced the desired signal (e.g., speech). At the same time, another sound source produced a disturbance (noise). In the experiments presented in this paper, both sources produced different speech signals. The sensor captured both signals mixed with each other. It was assumed that there is a sufficient degree of spatial separation between the sources, measured in the azimuth relative to the sensor.

The task of the algorithm is to extract the desired speech from the signals captured by the sensor, using the fact that the desired signal originates from a different direction than the disturbance. The algorithm was divided into two main parts. The first part used the four pressure signals to calculate sound intensity signals in X-Y directions and the source azimuth, in the frequency domain. The second part of the algorithm used the calculated azimuth values to select spectral components of the signal originating from the azimuth range of interest, and, then, it reconstructed the output signal. The details of the algorithm are presented in the subsections below.

### 2.1. Calculation of Sound Intensity

Sound intensity is a vector quantity describing the energy flow in sound waves, defined as the power carried by sound waves per unit area in a direction perpendicular to that area [22]. Sound intensity along an axis may be calculated using two closely spaced (c.a. 10 mm) omnidirectional microphones. If two pairs of microphones are placed on the orthogonal X-Y axes, so that the microphones are at an equal distance from the point where the axes cross, then a two-dimensional AVS is constructed. The proposed method analyzed the projection of sound intensity vectors on the X-Y plane. This approach provides correct results if all the sound sources are positioned at a comparable height. In case there are significant differences in the source height, a third orthogonal axis (Z) may be added to the sensor with an additional pair of microphones (a 3D AVS), so that both the azimuth and the elevation of sound sources can be analyzed. However, this approach was not needed for the experiments described in this paper. It is essential that the microphones are matched in terms of their characteristics; this may be ensured by performing a calibration process [28].

Four microphones of the sensor captured the pressure signals *p_x_*_1_, *p_x_*_2_, *p_y_*_1_, *p_y_*_2_. The average pressure *p* measured at the central point, located at an equal distance from all microphones, is equal to the following:(1)$$p\left(t\right)=\frac{p_{x1}\left(t\right)+p_{x2}\left(t\right)+p_{y1}\left(t\right)+p_{y2}(t)}{4}.$$

Acoustic velocity *u_a_* along the axis *a* is approximated with a pressure gradient:(2)$$u_{a}\left(t\right)≅p_{a1}\left(t\right)-p_{a2}\left(t\right),$$

Sound intensity *I_a_* along the axis *a* is given by the following [22]:(3)$$I_{a}\left(t\right)=\frac{1}{2\rho d}p_{a}\left(t\right)\int_{-\infty }^{t}u_{a}\left(\tau \right)d\tau ,$$
where *ρ* is the air density, and *d* is the distance between the pressure sensors.

Using the method described above, the total sound intensity was calculated. It is also possible to calculate sound intensity in the frequency domain. If *P_a_*(*ω*) and *U_a_*(*ω*) are the Fourier transforms of the average pressure and the acoustic velocity signals, respectively, then the sound intensity in the frequency domain is given by the following [12]:(4)$$I_{a}\left(\omega \right)=Re\left\{P_{a}(\omega )\cdot U_{a}^{*}(\omega )\right\},$$
where the asterisk denotes the complex conjugation. With this approach, sound intensity is calculated for the individual spectral components.

### 2.2. Spatial Filtration

The direction of the incoming sound (azimuth) is calculated for the individual frequency components of the signal, using the sound intensity spectra *I_x_*(*ω*) and *I_y_*(*ω*), obtained from the signals measured along the X and Y axes. The azimuth *φ*(*ω*) of the individual spectral components of the signal is given by the following:(5)$$\varphi \left(\omega \right)=arctan\frac{I_{x}(\omega )}{I_{y}(\omega )}.$$

In the experiments described in this paper, it was assumed that the direction of both sound sources is already known, and the spectral components having the azimuth within the defined range (*φ*_min_, *φ*_max_) originate from the source of a desired signal. Any spectral components with the azimuth outside this range were assumed to be a disturbance. Automatic detection of the number of active sound sources and their direction is possible, using the calculated intensity and azimuth values, but this is a separate problem which lies outside of the scope of this paper. Based on the azimuth, a binary mask *B*(*ω*) may be constructed:(6)$$B\left(\omega \right)=\left\{\begin{array}{l} 1,\;if\;\varphi_{min}\leq \varphi \left(\omega \right)<\varphi_{\max}\\ 0,\;otherwise \end{array},\right.$$
where *φ* is an unwrapped azimuth.

The processing algorithm operated on continuous, digital signals from the sensor. The actual processing was performed in the frequency domain. Therefore, the signal must be transformed using the Short-term Fourier Transform (STFT) [29]. The signal was processed in blocks of *N* samples, and the analysis window of length *N* was moved by *M* samples between the two consecutive blocks (the hop size). A block of samples with index *p* is given by the following:(7)$$x_{p}\left[n\right]=x\left[n+Mp\right]w\left[n\right],$$
where *x* is the processed signal, and *w*[*n*] is a windowing function, used to reduce the spectral leakage effect. Next, a discrete Fourier transform (e.g., FFT) was calculated:(8)$$S\left[q,p\right]=\sum_{n=0}^{N-1}x_{p}\left[n\right]exp\left\{-2j\pi qn/N\right\},$$
where *q* is the frequency bin index.

The spectrum calculated for a given block was then multiplied by mask *B* obtained for this block by the azimuth analysis according to Equation (6):(9)$$S^{\prime }\left[q,p\right]=S[q,p]\cdot B[q,p],$$
which sets the components *S*[*q*,*p*] having the azimuth outside the range of interest to zero.

The signal was then reconstructed by first performing the inverse discrete Fourier transform:(10)$$y_{p}[n]=\frac{1}{N}\sum_{q=0}^{N}S^{\prime }[q,p]exp\left\{2j\pi qn/N\right\},$$
and then by summing up the shifted blocks:(11)$$y\left[k\right]=\sum_{p}y_{p}\left[k-Mp\right]w_{d}\left[k-Mp\right],$$
where *w_d_*[*n*] is the canonical dual window of *w*[*n*].

In the experiments presented in this paper, a periodic von Hann window function was used:(12)$$w\left[n\right]=0.5\left(1-cos\left(2\pi \frac{n}{N}\right)\right),0\leq n<N.$$

In order to achieve accurate signal reconstruction with this type of window, the hop size *M* must satisfy the following condition:(13)$$M=\frac{N}{k+1},k=1,2,\ldots \;.$$

The dual window *w_d_* in Equation (11) was replaced by a constant value:(14)$$w_{d}\left[n\right]=\frac{2M}{N}$$

It was verified that this change does not affect the processing accuracy in a significant way.

Using the approach described above, a single output signal *y*[*k*], containing spectral components with azimuth from the defined range, was obtained. Multiple output channels, each with a different azimuth range of interest, may be obtained the same way.

## 3. Experiments and Results

### 3.1. Test Setup

An evaluation of the proposed algorithm was performed using a custom-built AVS. The sensor was constructed from miniature, omnidirectional, digital MEMS microphones, type ICS-43434 [30], connected to a computer through an I^2^S-USB interface. In order to ensure that the tests are performed in controllable and constant conditions, a set of impulse responses of the sensor for different positions of the sound source, was measured. The sensor was placed on a turntable in an anechoic room. The maximum length sequence (MLS) test signal was emitted from a stationary loudspeaker. For each source azimuth in the 0° to 355° range, with 5° resolution, impulse responses for all sensor channels were calculated by a correlation of the signals recorded from the sensor with the input MLS signal. Additionally, amplitude and phase correction functions were calculated to compensate for differences between the sensor microphones [28].

Four speech recordings, sampled at 48 kHz, with a duration of 70 s each, were selected for the evaluation of the proposed method: two female voice recordings (F1, F2) and two male voice recordings (M1, M2). The fundamental frequency of these voices, measured with the Praat software (version 6.1.09), were as follows: 195 Hz (F1), 194 Hz (F2), 124 Hz (M1) and 105 Hz (M2). The signals processed with the proposed algorithm were obtained by convolving the recordings with the sensor impulse responses for a selected azimuth. Two speech recordings were used in each test. The first speech recording was a desired signal (*x_s_*), always positioned at 0° azimuth. The second speech recording was a disturbance (*x_n_*), placed at various positions *φ**_n_* from 10° to 90°. The sensor is omnidirectional, and the exact positions of the sources are irrelevant; only the angular distance between the sources is important. For clarity of the presented results, the desired speech source was always positioned at 0°. A combination *x* = *x_s_* + *x_n_* was processed by the algorithm that divided it into two parts: *y_s_* and *y_n_*. In the ideal case, it is expected that *y_s_* = *x_s_* and *y_n_* = *x_n_*. There are six unique combinations of four voices (the combinations are symmetric, and it was verified that, e.g., variants M1-F1 and F1-M1 yield consistent results). Of these six variants, three representative ones will be evaluated in this paper: M1-F1, M1-M2 and F1-F2 (for brevity, they will be labeled as M-F, M-M and F-F, respectively, throughout the text). The remaining three variants yielded similar results (e.g., M1-F1 and M1-F2), so they are omitted from the presented results for clarity.

The simulations were performed on a computer. A speech signal combined with the disturbance was divided into blocks of *N* = 4096 samples, with the window hop size equal to *M* = 64 samples (the choice of these values is discussed further in this paper). For each block of samples, intensity signals were calculated, and the azimuth of spectral components was determined. The mask *B*(*ω*) was calculated by assigning spectral components with azimuth in the range (−*φ**_n_*/2, *φ**_n_*/2) to *y_s_* and the remaining ones to *y_n_*. The choice of the azimuth range is explained further in this paper. Next, each block of the input signal was multiplied by a von Hann window, a Fast Fourier Transform (FFT) was computed and multiplied by the mask *B*(*ω*), an inverse FFT was calculated, and the result was added to the buffer. The first *M* samples were then sent to the output, and the remaining part of the buffer was right padded with *M* zeroes. This procedure was repeated for all the consecutive sample blocks.

### 3.2. Examples of the Processing Results

Figure 2 shows example spectrograms of the input signals and the processing result (a male speech at 0° mixed with a female speech at 30°). It can be observed that spectral components that were present in the original speech signal are retained, while the components introduced by the disturbance are removed from the algorithm output.

Figure 3 shows an example mask *B*(*ω*) of spectral components created by the algorithm during the processing. The black color indicates the spectral components that are marked for removal, as their azimuth is outside the range of interest. Short black horizontal lines may be observed for low frequencies—these are the strongest components of the disturbance (formants in the spectrum of speech from the second source). The white color denotes the components that are retained.

### 3.3. Efficiency of Spatial Filtration

A signal-to-distortion ratio (SDR) is a standard metric used for the evaluation of separation algorithm efficiency [31]. The SDR coefficient expresses the ratio of energy of the target signal (the desired signal in the separation algorithm output) to the total energy of the interference (the remaining signal from the disturbance source), processing artifacts and noise. The SDR value is presented in decibels, and a higher value means better efficiency of the separation algorithm.

The results obtained for different positions *φ**_n_* of the disturbance signal *x_n_* source are presented in Figure 4. The desired signal source was always at 0°. Both input signals had an equal rms (root-mean-square) value, so the input signal-to-noise ratio (SNR) was 0 dB. It can be observed that the obtained SDR values are above 9 dB even for a small distance between the sources (≥15°). As the distance increases, the SDR becomes larger, although the differences are small. This confirms that the proposed method successfully separates the signals even if the azimuth difference between the sources is small (15° is sufficient). As expected, the SDR is the highest for the M-F case, because the overlap between the spectral components from the male and the female voices is smaller than in the M-M and F-F cases. The SDR for the M-M case is lower than for the F-F case; this may be caused by the fact that the overlapping spectral components from the male speakers had higher energy. Nevertheless, the algorithm was able to separate the voices successfully even if two speakers of the same sex were active simultaneously.

### 3.4. Selection of the Azimuth Range of Interest

In the experiment presented in the previous section, the range of the azimuth of the spectral components assigned to *y_s_* was equal to ±*φ_n_*/2, where *φ_n_* is the azimuth of the source of the disturbing signal *x_n_*. The azimuth range is therefore not constant, as it depends on the position of the noise source. Figure 5 presents the SDR values calculated for various azimuth ranges ±*φ* for the M-F case. In this experiment, the signal source was placed at 0°, the noise source was placed at 90°, and the rms of both signals were equal.

From the results, it can be observed that the SDR value initially increases with the azimuth range, reaching a plateau for 40–60°, and, then, it starts to decrease. Therefore, the optimal azimuth range is between ±40° and ±60°, and the explanation for this is as follows: If a spectral component is present in both the signal and the disturbance, the observed azimuth of this component in the combined signal is a result of superposition of these two sources. For example, if two spectral components of the same amplitude and frequency are emitted by two sources at 0° and 90°, the result will be a component with the azimuth equal to 45°. Therefore, preserving all spectral components of *x_n_* that are stronger than the components of *x_n_* requires the azimuth range of interest equal to approximately ±*φ_n_*/2. During the initial experiments, a constant azimuth range of ±10° was used, and this approach led to poor separation results. In the experiments presented in this paper, the azimuth range was ±*φ_n_*/2.

The discussion presented above is valid for the input SNR equal to zero. If the signal is stronger than the disturbance (SNR > 0), a narrower azimuth range may be sufficient. Similarly, if the disturbance is stronger than the signal (SNR < 0), a wider range of azimuth may be required. In practical applications, the parameter defining the azimuth range should be tunable.

### 3.5. Speech Intelligibility

The important question is whether the spatial separation of sound sources with the proposed algorithm improves speech intelligibility over the unprocessed signal case. A Short-Time Objective Intelligibility Measure (STOI) was proposed by Tal et al. [32] for the evaluation of methods where noisy speech is processed by time–frequency weighting, such as noise reduction and speech separation. The STOI metric is intrusive, i.e., it is calculated by comparing clean speech with degraded speech. A frequency range up to 5 kHz is analyzed. The test result is a value in the range 0 to 1, where a higher value indicates better speech intelligibility.

Table 1 shows the STOI values calculated by comparing the clean signal *x_s_* with a degraded, unprocessed signal *x_s_* + *x_n_*, which serves as a baseline for comparison, and with the selected processed signals *y_s_*. Figure 6 presents the results for all tested values of the source separation *φ_n_* (*x_s_* source was always positioned at 0°), expressed as a difference between the STOI values obtained for the signal processed by the algorithm and the distorted input signal. The remaining parameters are the same as in the previous test.

From the presented results, it may be concluded that the proposed method provides a substantial improvement in speech intelligibility when the two simultaneously active sound sources are separated. If the angular distance between the sources is at least 15°, the resulting STOI scores exceed 0.875. In all cases, the STOI difference between the processed and the unprocessed signal increases slightly with the distance between the sources. The largest increase in the STOI is obtained for the M-F case, and the smallest for the F-F case. However, the STOI calculated for the unprocessed F-F case was the highest of all three values. The algorithm provided the smallest STOI gain in this case (although an increase of 0.1 is significant), but the absolute STOI results for the processed F-F case are the highest of all three. At the same time, the unprocessed STOI for the M-F case was the lowest, and the algorithm provided a largest STOI increase for this case. Therefore, the proposed algorithm is most beneficial in terms of speech intelligibility if the initial speech intelligibility for the desired speech mixed with a disturbance is low.

### 3.6. Speech Quality

Speech quality is a measure of all audible distortions in the signal, perceived by the listener, independently of their influence on speech intelligibility. A Perceptual Evaluation of Speech Quality (PESQ) measure, described in ITU-T standard P.862 [33] was used to evaluate the degree of audible distortions in the processed signal. The PESQ metric is also intrusive; it is calculated by comparing the degraded speech with a clean speech. The test was performed in a wide-band mode, with frequencies up to 8 kHz being analyzed. The PESQ result is a value on a mean opinion score (MOS) scale, with values from 1.0 (the worst quality) to 4.5 (the best quality).

Table 2 shows the PESQ values calculated by comparing the clean signal *x_s_* with a degraded, unprocessed signal *x_s_* + *x_n_*, which serves as a baseline for comparison, and with the selected processed signals *y_s_*. Figure 7 shows the PESQ scores calculated for all tested values of the source separation *φ_n_* using the same parameters as in the STOI test.

The obtained PESQ values are low (1.8–2.1), corresponding to the MOS grades Bad/Poor. This indicates that the processed signals are strongly degraded. Still, a substantial increase in the PESQ over the unprocessed signals is obtained with the algorithm. As the STOI results indicate, the degraded quality does not significantly affect speech intelligibility. The audible distortions have a form of signal modulation, which is a result of the removal of spectral components from the signal. The algorithm works by selecting the individual spectral components of the signal, based on their azimuth. If a component is constantly selected or removed, it results in a modulation distortion of the signal. Such an effect is common to many speech processing algorithms, such as active noise reduction or adaptative filtering. Moreover, setting the spectral components of the signal to zero is an equivalent to ‘brick filtering’ in the frequency domain, which introduces additional signal distortion. Suppression of these types of distortion and increasing the speech quality is a separate topic that is left for future research. It should also be noted that, if the proposed algorithm is used as an input to an automatic speech recognition system, speech quality is not an issue.

### 3.7. Influence of the Input Signal-to-Noise Ratio on the Results

The experiments presented so far have been performed for the 0 dB SNR at the input; i.e., the rms values of the desired signal and the disturbance were the same. It is important to test the algorithm performance in case of a varying input SNR. The test was performed for the angular distance between the sources equal to 45°. Figure 8 presents the SDR values obtained for different input SNR values. Figure 9 shows the calculated STOI values and their differences between the processed and the unprocessed signals.

As expected, SDR and STOI values increase when the input SNR becomes larger. All three cases reach the STI range 0.98–0.99 for 15 dB of the SNR. Differences in the STOI between the three cases became smaller as the SNR increased, while the differences in the SDR remained approximately constant. The gain in the STOI after the processing is the largest for a negative SNR, i.e., when the disturbance is stronger than the desired signal. As the SNR increases, the STOI difference becomes smaller. These results indicate that the proposed algorithm is most beneficial when the desired signal is strongly masked by the disturbance (SNR ≤ 0 dB).

It was explained earlier in this paper that, if two sound sources are active at the same time, a superposition of these sources occurs, and the observed azimuth lies between the actual sources. For a SNR > 0 dB, the signal components are stronger than the noise components, so the resulting component is closer to the signal source position. The opposite effect is observed when the SNR < 0 dB: the resulting component is closer to the noise source position. If the azimuth range defined for the processed signal in the algorithm is sufficiently large (as the one used in the experiments), an efficient signal separation is possible even for a small SNR.

### 3.8. Choice of the Block Size and the Hop Size

The proposed algorithm is based on time–frequency signal processing. The input signal is divided into blocks of *N* samples. Each block is multiplied by a window function and transformed into the frequency domain with the Fast Fourier Transform. The signal is processed and then reconstructed. The following experiment evaluated the algorithm performance for various choices of the Fourier transform size *N* and the hop size *M*. For performance reasons, *N* is assumed to be a power of two. The hop size *M* must satisfy Equation (13). Table 3 presents the SDR values calculated for the M-F case. The other two cases (M-M, F-F) yielded the results consistent with the presented one. The presented values are valid for a signal sampled at 48 kHz. The angular distance between the sound sources was constant and equal to 90°, and the input SNR was 0 dB.

A larger block size *N* is required to achieve good frequency resolution of the processing, but, at the same time, larger *N* deteriorates the temporal resolution and increases the processing time. In this case, a smaller hop size *M* (a larger overlap between the blocks) is required to improve the temporal resolution, at a cost of increasing the processing time even more. From the results presented in Table 3, it can be observed that the highest SDR was obtained for *N* = 4096 or 2048 and for *M* ≤ *N*/16. Therefore, the authors used *N* = 4096 and *M* = 64 throughout all the experiments. However, if the processing time is an issue (e.g., in real time applications), a choice of *N* = 2048 and *M* = 128 provides a satisfactory SDR value, above 12 dB. A small block size (*N* = 1024) is not sufficient to achieve the necessary frequency resolution of the processing. Larger block sizes (e.g., *N* = 8192) produced worse results than the shorter blocks because of the temporal processing resolution being too low, and the processing time was unnecessarily increased.

### 3.9. Speech Recognition Accuracy

To conclude the experiments, a test of automatic speech recognition with both the unprocessed and the processed speech signals was performed. The recordings were processed with the Google speech-to-text system, using its web interface [34]. The converted text was compared with the ground truth data, and two performance scores were calculated for each case: a word error rate (WER) and a character error rate (CER) [35], using the Huggin Face metric evaluation tool [36]. The results are presented in Table 4 (WER) and Table 5 (CER). Higher values of both metrics indicate worse recognition accuracy (a higher ratio of errors). The reference values were obtained for the clean speech recordings (M1 and F1). The unprocessed results were calculated for the speech recordings mixed with the interference. The remaining three values were obtained for the signals processed with the proposed algorithm and for the three values of the angular distance between the sources.

As expected, the ASR system performed poorly with the unprocessed mix of the desired speech with the interference, resulting in WER ≥ 50% and CER > 38%. After processing the distorted speech using the proposed algorithm, both metrics have improved considerably. The resulting WER scores were larger by 1.8–3.5% than the reference, and the resulting CER scores were larger by 0.3–1.8% than the reference.

## 4. Discussion

### 4.1. Evaluation of the Algorithm Performance

The proposed algorithm for the spatial separation of signals from simultaneously active sources, based on the sound intensity analysis, was evaluated in a series of experiments. The overall results indicate that the algorithm works correctly. A speech signal of interest is mixed with another speech signal and treated as a disturbance, and the mixed signal is processed with the algorithm. The processing result contains the original speech signal without the added disturbance, thus achieving the spatial separation effect. This was possible thanks to processing the signal captured by the AVS, consisting of four closely spaced omnidirectional microphones, performing sound intensity analysis, determining the azimuth of the individual spectral components, selecting only components that originate from a specific direction, and finally reconstructing the signal. This approach differentiates the proposed method from the related ones, which are based on processing pressure signals from the individual microphones.

The algorithm was evaluated by calculating three metrics: SDR (separation efficiency), STOI (speech intelligibility) and PESQ (speech quality). Each metric was computed by a comparison of the processing results with the clean (desired) speech signals. All three metrics confirmed that the proposed algorithm significantly reduces the interference from the unwanted speech source. The algorithm improves speech intelligibility over the unprocessed signal case, achieving high STOI values (>0.87) in all tested cases. This aspect is especially important if the algorithm is to be used as a preprocessor for an ASR system. The authors confirmed this assumption with a test performed using the Google Web Speech ASR. The WER and CER scores obtained for the processed signals were significantly lower (which means a higher accuracy) than for the unprocessed signals, and in the same order as the scores for the original speech recordings. The maximum increase in the error rate over the reference was 3.5% for WER and 1.8% for CER. The results of this experiment confirm that the proposed algorithm processes speech signals in a way that an ASR accuracy is improved in case of concurrent speech. It also confirms that low PESQ scores obtained for the processed signals do not influence the ASR system performance in a significant way.

The authors presented three cases of female and male speech recordings mixed with each other. Some other cases were also evaluated, and they yielded results consistent with these presented in this paper. There are differences in the results obtained for different cases, but a significant improvement in each metric was observed in each case. The results also indicate that, although the metrics improve as the angular distance between the source positions increases, the differences are not as large as it was expected. In fact, a minimum separation between the sources is only 15°, and it is sufficient to obtain a satisfactory accuracy of the algorithm. As discussed before, the largest improvement in all metrics is obtained if the input SNR is low (≤0 dB). If the distance between the sources is too low (<15°), the azimuth ranges of the signal components originating from each source overlap in a way that it is not possible to reliably decide which source the components originate from. This effect causes a rapid decrease in all metrics when the distance between the sources is small.

One aspect that leaves room for improvement in future research is speech quality. This is not important for the ASR systems, but if the processed speech is to be presented to a listener, speech quality in the current algorithm version is acceptable, but it should be improved. As discussed earlier, the main source of the quality deterioration is setting the spectral component of the signal that lies outside of the azimuth range of interest, to zero. This causes an audible modulation of the sound. A better approach is to ‘fill’ the spectral components that were removed. For example, a prediction filter may be applied for the task, interpolation in the frequency domain may be used for small gaps in the spectrum, or both methods may be used together. An alternative approach is to use a trained neural network to ‘guess’ the values of the missing spectral components, based on the two-dimensional (temporal and spectral) analysis of the signal. These improvements are outside of the scope of this paper, and they will be researched in the future.

This paper did not address the problem of detecting the source position (azimuth), as this is a separate issue that requires separate research. The presented experiments were focused on the evaluation of the algorithm processing the degraded signals. To make this evaluation possible, sound source positions were fixed in the simulation. In a practical system, a detector of sound sources is needed. The proposed algorithm provides input data for such a detector, i.e., the azimuth of all spectral components. Statistical analysis of these data collected from several signal blocks may be performed, e.g., by computing an azimuth histogram. Then, a decision system must be developed to select important sound sources and define the azimuth range of each of them, passing these data to the algorithm presented here. The problem of a successful detection of simultaneous sound sources is that the source superposition occurs, as described earlier in this paper. Therefore, the detection problem was left for future research.

### 4.2. Comparison with the Related Methods

Performing a meaningful comparison with state-of-the-art methods is difficult, as various researchers used different recordings and processing scenarios (not all the published results were obtained for the concurrent speech), and they report different metrics, often without providing the baseline values for the distorted speech. In Table 6, the results obtained with the proposed method are compared with the values reported in the related publications. The description of the methods presented in Table 6 is provided in the Introduction section of this paper. The STOI results from the proposed method are comparable with the ones reported by other authors. The SDR scores obtained with the algorithm presented in this paper are comparable with the results from methods not based on machine learning. The obtained PESQ values are slightly lower. However, for the ASR systems, the quality measure (PESQ) is less important than the intelligibility score (STOI).

### 4.3. Limitations of the Algorithm and Future Work

The main limitation of the proposed method is that positions of sound sources must either be known a priori or they must be determined with signal analysis. In the experiments presented in this paper, for the purpose of the algorithm evaluation, the positions of the sources were constant and predetermined. However, it is possible to perform an analysis of the azimuth values calculated by the algorithm and to detect the source position automatically. Also, in real world applications, it cannot be assumed that the positions of the sound sources are constant. Therefore, the algorithm must be supplemented with a source tracking method, which will be the topic of future research.

Another problem is the presence of reflected sound waves in real acoustic environments. In the experiments described in this paper, the authors focused on evaluating the algorithm performance in the controlled conditions, so the reverberation was neglected. In real acoustic spaces, it may be expected that the separation accuracy may deteriorate if a large amount of energy of the reflected sound waves will be received by the sensor from the same direction as the desired signal. The degree of this deterioration will depend on the room configuration and the acoustic properties of reflecting surfaces. However, in the case of concurrently active sound sources, these reflections will usually be masked by a stronger direct signal, and, if only one source is active at a time, the reflections may be suppressed with additional signal processing, e.g., using adaptive filters. The authors performed preliminary, informal experiments in real rooms with the proposed algorithm and an automatic speech recognition system, and it was observed that the separation provided by the proposed algorithm improves the accuracy of the recognition system.

Another limitation of the proposed method is that it requires a specific multichannel sensor (AVS), while the methods based on machine learning operate on a monophonic sound recorded with a single microphone. However, the experiments presented in this paper were performed employing a single, low-cost sensor. Finally, as stated earlier in this paper, the limitation is that there must be a sufficient angular distance between the sound sources (>15°) to ensure a proper performance of the separation algorithm.

## 5. Conclusions

In this paper, an algorithm for separation of the speech signal from the simultaneous disturbance was presented and evaluated. The algorithm works on sound intensity signals obtained from an acoustic vector sensor, which is a different approach from most of the state-of-the-art methods based on the pressure signals. The performed experiments proved that the proposed method achieves good accuracy of the spatial signal separation signals, as long as the angular distance between the sources is at least 15°. In the experiments performed with the optimized algorithm parameters, the algorithm provided SDR values 10–12 dB, improvement in the STOI by 0.15–0.30, and improvement in the PESQ by 0.61–1.00. The achieved level of speech intelligibility (STOI) is 0.86–0.94 if the input SNR is 0 dB, and an SNR ≥ −15 dB is needed to obtain an STOI exceeding 0.65. The algorithm works correctly for various types of speech signals (female or male speech). The optimal size of the analysis window for signals sampled at 48 kHz is 2048 samples (42.67 ms), with a hop size of 128 samples (93.75% overlap).

The presented method is intended to be used as a preprocessing algorithm for the automatic speech recognition systems, for the reduction in interference introduced by the concurrent speakers. The state-of-the-art methods for concurrent speech separation can be divided into two groups. The single channel methods are usually based on the machine learning approach, requiring a large training set and powerful hardware to run. Most of the multichannel methods are based on large, distributed sensor arrays that are impractical for real-world applications. The approach presented in this paper uses a small-size sensor, the algorithm can be run on a simple microcomputer in real time, and the speech separation efficiency is comparable with the state-of-the-art methods.

The future research will focus on improving the processed speech quality (reducing the distortion introduced by the processing) and on the automatic detection of active sound sources. The experiments presented in this paper were obtained in a simulated and controlled environment, which allowed for evaluation of the proposed algorithm. In the next stage of research, the algorithm will be tested in real acoustic conditions, with various speakers, in varying acoustic conditions (different rooms, various types of the disturbance—environmental noise, music, etc.). Preliminary tests were conducted, and it was found that, while the accuracy of source separation is lower than in the simulated conditions (mostly due to the presence of reverberation), the overall performance of the algorithm is satisfactory and sufficient for speech recognition applications. In future research, the authors plan to integrate the proposed method within an automated speech recognition system.

## Figures and Tables

**Figure 1 sensors-25-01509-f001:**
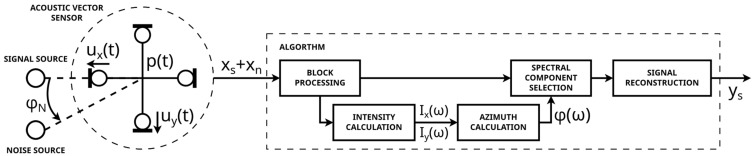
Block diagram of the proposed algorithm for the separation of the desired signal from the disturbance.

**Figure 2 sensors-25-01509-f002:**
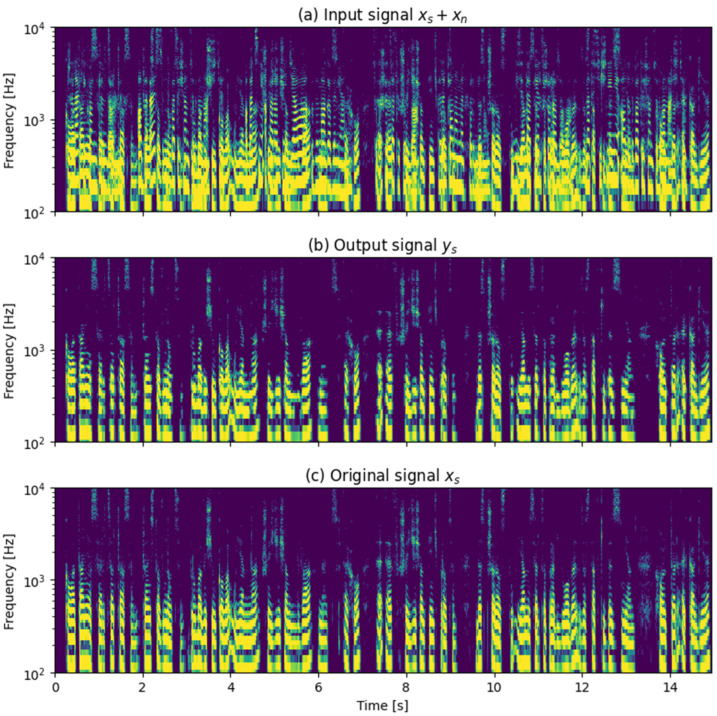
Example spectrograms: (**a**) input signal to the algorithm—male speech *x_s_* at 0° mixed with the disturbance *x_n_* (female speech) at 30°, (**b**) the processing result *y_s_*, and (**c**) the original signal *x_s_*. Color represents the spectral magnitude level in dB.

**Figure 3 sensors-25-01509-f003:**
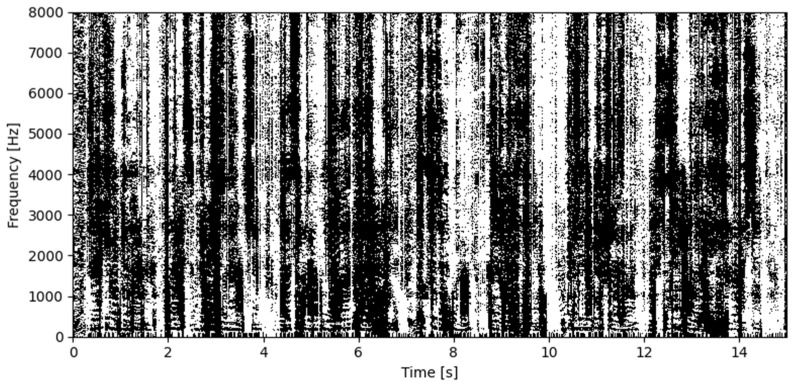
Example of the signal processing: a binary spectral mask. Black color denotes spectral components that are removed from the signal.

**Figure 4 sensors-25-01509-f004:**
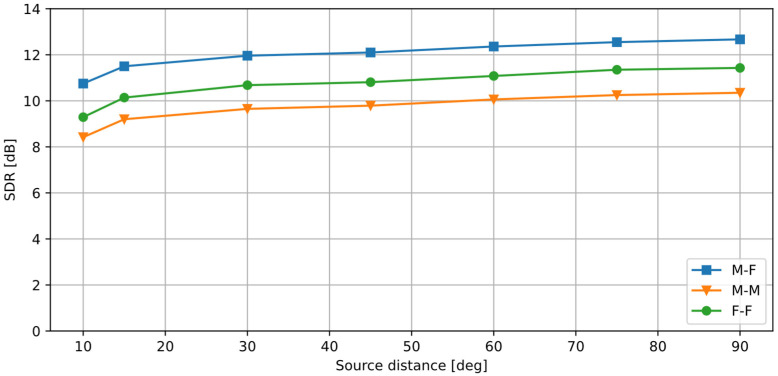
SDR values calculated between the input *x_s_* and the output *y_s_* signals, for a varying angular distance between the sources. Three combinations of female (F) and male (M) voices are presented.

**Figure 5 sensors-25-01509-f005:**
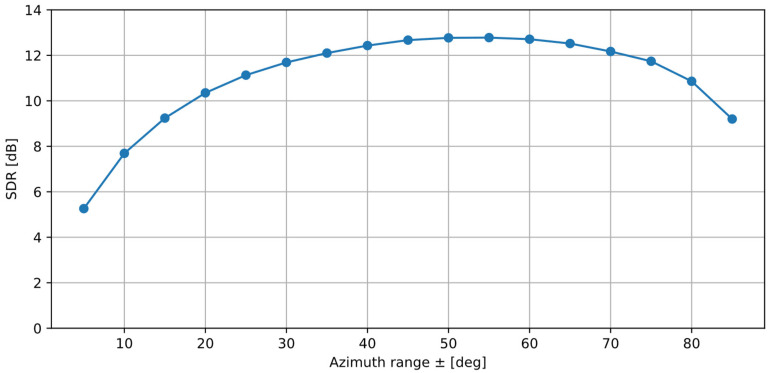
SDR values calculated for a varying azimuth range of interest for the desired signal for the M-F case (M source at 0°, F source at 90°, equal SNR).

**Figure 6 sensors-25-01509-f006:**
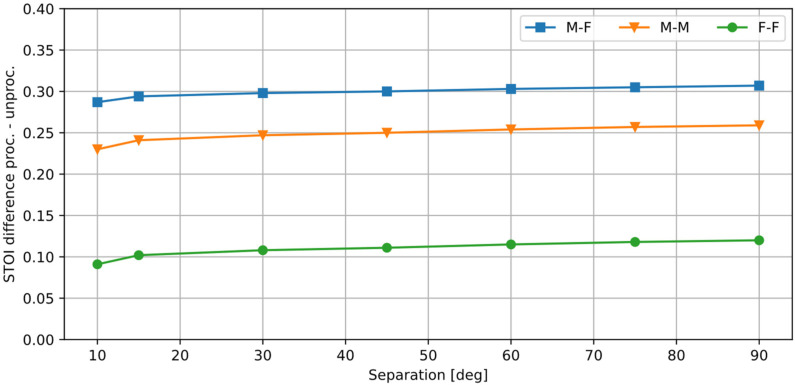
STOI difference between the processed and the unprocessed signal, for a varying angular distance between the sources.

**Figure 7 sensors-25-01509-f007:**
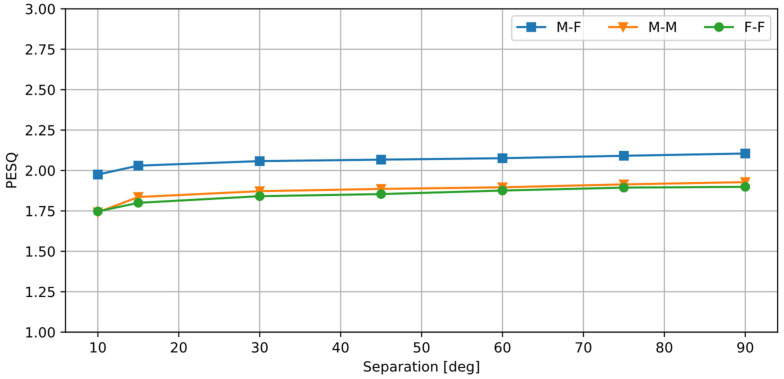
PESQ values calculated for a varying angular distance between the sources.

**Figure 8 sensors-25-01509-f008:**
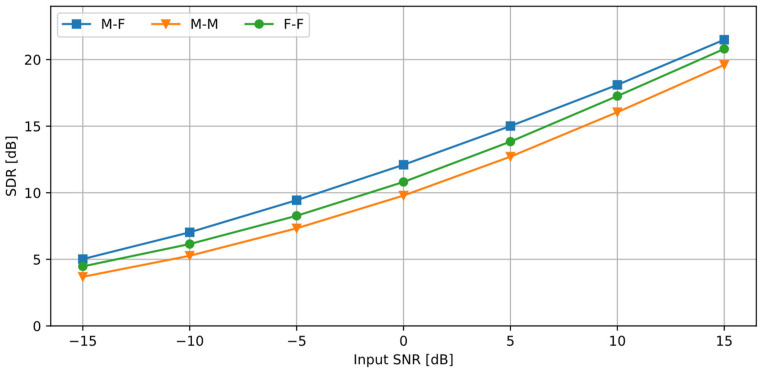
SDR values calculated for a varying input SNR (source distance = 45°).

**Figure 9 sensors-25-01509-f009:**
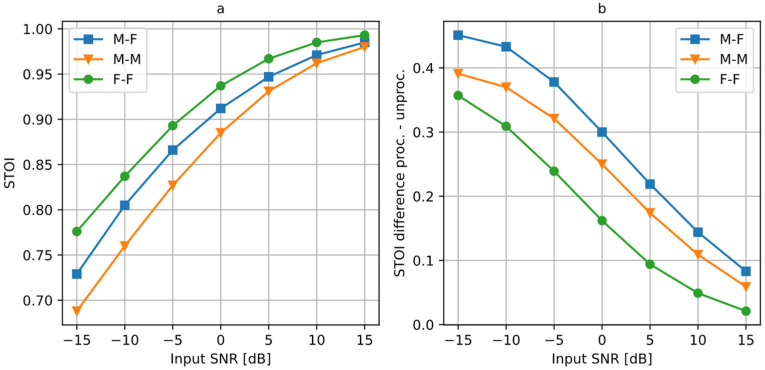
STOI results calculated for a varying input SNR (source distance = 45°): (**a**) STOI values, and (**b**) the difference between the STOI for the processed and the unprocessed signal.

**Table 1 sensors-25-01509-t001:** STOI scores for the unprocessed signals and for the selected processed signals.

Case	Unproc.	15°	45°	90°
M-F	0.612	0.906	0.912	0.919
M-M	0.635	0.876	0.885	0.894
F-F	0.774	0.929	0.937	0.945

**Table 2 sensors-25-01509-t002:** PESQ scores for the unprocessed signals and for the selected processed signals.

Case	Unproc.	15°	45°	90°
M-F	1.101	2.030	2.058	2.105
M-M	1.127	1.836	1.872	1.928
F-F	1.115	1.872	1.841	1.899

**Table 3 sensors-25-01509-t003:** SDR values calculated for a varying block size *N* and hop size *M*, for the M-F case, and for the angular distance between the sources equal to 90°.

Hop Size M	Block Size N			
	1024	2048	4096	8192
64	9.90	12.08	12.67	11.39
128	9.92	12.04	12.66	11.39
256	9.54	11.92	12.63	11.38
512	8.75	11.63	12.54	11.35
1024	–	10.64	12.32	11.27
2048	–	–	11.21	11.09
4096	–	–	–	10.34

**Table 4 sensors-25-01509-t004:** WER scores [%] for the original speech recordings (Reference), speech mixed with the interference (Unproc.), and the processing results for three positions of the interference source.

Case	Reference	Unproc.	15°	45°	90°
M-F	3.5	50.0	6.1	6.1	7.0
M-M	3.5	81.6	6.1	5.3	6.1
F-F	3.2	74.5	5.3	6.4	5.3

**Table 5 sensors-25-01509-t005:** CER scores [%] for the original speech recordings (Reference), speech mixed with the interference (Unproc.), and the processing results for three positions of the interference source.

Case	Reference	Unproc.	15°	45°	90°
M-F	1.8	38.3	2.6	2.6	3.7
M-M	1.8	61.6	2.6	2.2	2.1
F-F	0.6	39.1	1.0	1.2	1.0

**Table 6 sensors-25-01509-t006:** Comparison of the results obtained with the proposed method with the values reported by other researchers.

Source	SDR [dB]	STOI		PESQ	
	Processed	Original	Processed	Original	Processed
Proposed method	8.42–12.67	0.67	0.90–0.92	1.11	1.91–1.98
Deep learning monoaural separation (review) [13]	6.7–22.3	n/a	0.88–0.98	n/a	2.03–2.83
Time-dilated convolutional network + U-net [10]	n/a	0.59	0.81	1.55	2.38
Multi-modal separation [19]	14.2–15.5	0.67	0.92	1.17	2.41
Neural beamformer (Convolutional Neural Network) [27]	3.79	n/a	0.59	n/a	1.76
Initial-iterative neural beamformer [18]	22.31–24.60	n/a	0.99	n/a	4.00–4.25
Expectation-maximization on B-format signals [21]	3–12	n/a	n/a	n/a	2.00–2.26

n/a: information was not provided by the authors.

## Data Availability

Dataset available on request from the authors.

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
