# Peer review of "Separation of Simultaneous Speakers with Acoustic Vector Sensor"

_sensors, 2025, doi:10.3390/s25051509_

Round 1

Reviewer 1 Report

Comments and Suggestions for Authors

The paper presents a method of separating a speaker voice from noise, which is modeled by adding one more speaker. The method is based on subsequently done: Fourier spectral decomposition, azimuth determination for individual spectral components, masking components with improper azimuth, inverse Fourier transform. The idea is clear and seems working.

However some important details are not discussed. First, the authors consider speaker and noise (i.e. another speaker) are located in OXY plane of the microphone. This may be easily violated in real life. What is the tolerance of the method against positioning of the speaker and/or noise out of OXY plane. Please provide some elaboration.

Second, it is claimed the speaker is located along OX axis. Again, this might not be true in applications. Analogously, please provide some elaboration. 

Beginning part of the introduction lacks citations. There are many facts discussed, that should be supported by referring to appropriate sources.

Smaller issues.

  1. Line 174 contains P_a(\omega) and U_a(\omega). Please provide formulas.
  2. Formula 3. Letter 't' is upper limit of integration and meanwhile the integration variable. This is not correct. Please use different letter for integration variable.
  3. Abbreviations should not be used (or should be disclosed) in the abstract.
  4. Formulas for SDR, STOI and PESQ would improve the readability.

Reviewer 2 Report

Comments and Suggestions for Authors

A brief summary:

This paper focuses on the problem of sound source separation in real-time audio signals and proposes an innovative method based on sound intensity analysis. It uses a small acoustic vector sensor to collect sound pressure signals and calculate the two-dimensional sound intensity spectral distribution, and then filters the signal spectral components according to the sound source direction to achieve spatial filtering.

Specific comments:

  1. Please describe the content of the dataset used in more detail. For example, present it in a table, including the number of voices used in training and the number of voices in testing. The variable number of human voices has always been an issue.
  2. There is a lack of a "Limitations" chapter to describe the shortcomings of this paper.
  3. The proposed method should be compared with multiple methods, such as the classic TasNet, TDANet, and the 2024 TDFNet, SepReformer, etc. Nowadays, most methods are based on the Transformer architecture, while the proposed method still relies on Fourier transform.
  4. In actual complex acoustic environments, such as when there is a large amount of echo and dynamically changing background noise, to what extent will the sound intensity analysis and azimuth determination mechanism on which this method depends be interfered? Are there any potential improvement strategies to deal with such complex situations?
  5. The sensor calibration process mentioned in the paper is only briefly described. Then, in the case of changes in environmental factors such as different temperatures and humidities, how is the performance stability of the sensor? Will these environmental factors significantly affect the sound intensity calculation and the final sound source separation accuracy?
  6. For the non-stationary characteristics existing in the speech signal, such as sudden volume changes, speech rate changes, and the unique pronunciation habits and prosodic features among different speakers, does this method have sufficient adaptability in dealing with these complex situations? Has there been a detailed experimental analysis and evaluation of these factors, and how to ensure high separation performance in these cases?
  7. When multiple sound sources are very close in space and the angular difference is extremely small (less than the minimum 15° mentioned in the paper), the performance of the algorithm will decline sharply. The paper only mentions this phenomenon but does not deeply explore the underlying mathematical principles. Can the fundamental reason for this performance decline be deeply analyzed from the perspective of signal processing and acoustic theory, and possible theoretical improvement directions be proposed?
Comments on the Quality of English Language

The English could be improved to more clearly express the research.

Reviewer 3 Report

Comments and Suggestions for Authors

This paper presents a method of sound source separation in live audio signals, based on
sound intensity analysis. 

The topic is interesting and well-presented. Also, the quality of the figures is good and the references are adequate. However, I have some comments:

1- One sound source produces the desired signal (e.g. speech). At the same time, another sound source produces a disturbance (noise), so I assume you can do a simple speech enhancement not separation is that correct? Otherwise, the word noise is quite confusing.

2- Please check in the text, you use Figure and Fig. Please unify the notation.

3- On page 16, the paragraph starts with "Performing a meaningful comparison with the state-of-the-art methods is difficult". It is quite complicated to follow please modify it, and make it as a table.

4- I understand it is not fair to compare your approach with others because your recordings were different, but you can use SOTA approaches on your recordings and show the results for fair comparison, is it possible? 

Round 2

Reviewer 1 Report

Comments and Suggestions for Authors

The author have addressed the comments of the previous review and modified the paper accordingly. The work can be published now.

Author Response

We thank the Reviewer for the analysis of our manuscript and for the positive recommendation.

Reviewer 2 Report

Comments and Suggestions for Authors

The authors have addressed all the questions. However, in Table 6, the baseline methods should use method names instead of authors’ names to improve clarity (e.g., “BERT-based Framework” rather than “Chen et al. [10]”). Additionally, these comparative methods ought to be introduced individually with technical descriptions, such as their core algorithms, application contexts, and implementation specifics, to help readers better understand their roles in the evaluation.

Comments on the Quality of English Language

The English could be improved to more clearly express the research.

Author Response

Table 6 was modified according to the Reviewer's suggestion. All the SoA methods presented in the table are briefly described in the SoA review in the Introduction section, a comment was added to the paper to indicate that. An interested reader may find the details of these complex methods in the cited publications.